# Analysis of Chemical and Biochemical Parameters of Petrol-Contaminated Soil after Biostimulation with an Enzyme Reagent

**Kornel Curyło [1], Arkadiusz Telesiński [1,\*], Grzegorz Jarnuszewski [2],
Teresa Krzyśko-Łupicka [3] and Krystyna Cybulska [1]**

[1] Department of Bioingineering, Faculty of Environmental Management and Agriculture, West Pomeranian University of Technology in Szczecin, 17 Słowackiego St., 71-434 Szczecin, Poland; kornel.curylo@zut.edu.pl (K.C.); krystyna.cybulska@zut.edu.pl (K.C.)

[2] Department of Environmental Management, Faculty of Environmental Management and Agriculture, West Pomeranian University of Technology in Szczecin, 17 Słowackiego St., 71-434 Szczecin, Poland; grzegorz.jarnuszewski@zut.edu.pl

[3] Independent Department of Biotechnology and Molecular Biology, Faculty of Natural Sciences and Technology, University of Opole, 6a Kardynała Kominka St., 45-035 Opole, Poland; teresak@uni.opole.pl

\* Correspondence: arkadiusz.telesinski@zut.edu.pl

**Abstract:** This study aimed to assess the effect of petrol and the Fyre Zyme reagent on selected chemical and biochemical properties of loamy sand. The experiment was conducted under laboratory conditions. First, petrol was introduced into the soil at doses of 0 and 50 g k$^{-1}$dry matter (DM). Next, 6% Fyre-Zyme enzyme reagent solution was added to the samples contaminated and uncontaminated with petrol, in the following combinations: 0 (control), once at 40 cm$^3$ kg$^{-1}$ DM, twice at 20 cm$^3$ kg$^{-1}$ DM at two-week intervals, and four times at 10 cm$^3$ kg$^{-1}$ DM at weekly intervals. Contamination of loamy sand with petrol caused slight changes in the determined chemical parameters and stimulated dehydrogenase activities, but inhibited the activity of phosphatases. The introduction of the enzyme reagent into the soil increased the C$_{org}$ and N$_{tot}$ content. The greatest changes were observed in the activity of phosphatases. The receiver operating characteristic (ROC) curves revealed that the application of the enzyme reagent at the application of $4 \times 10$ cm$^3$ kg$^{-1}$ DM was the most beneficial. However, the results of the $\eta^2$ analysis indicate that the greatest influence on the determined experimental parameters was found in the soil contaminated with petrol.

**Keywords:** dehydrogenases; Fyre-Zyme; hydrocarbons; nitrogen total; organic carbon; phosphatases; soil remediation

## 1. Introduction

Petrol is a transparent, oil-derived liquid, used primarily as a fuel in combustion engines. It mainly consists of organic compounds formed from the fractional distillation of crude oil, enriched with various additives [1]. These compounds include aliphatic hydrocarbons, with 6–12 carbon atoms in a chain. Petrol also contains some amounts of aromatic hydrocarbons and unsaturated hydrocarbons, but they form only a minor part of the composition [2]. After penetrating the soil in significant amounts, these hydrocarbons may cause clumping and change the physicochemical and biological properties of soil. Moreover, light petrol fractions are mobile, reactive, and highly toxic to soil microorganisms at high concentrations (of the order of several hundred mg kg$^{-1}$) [3]. The bioavailability of aliphatic and aromatic hydrocarbons to microbial cells depends on physical (soil structure, pore size), chemical (adsorption, ion exchange, complexing), and microbiological (mainly cell wall structure) factors [4].

The decomposition of hydrocarbons in the soil environment occurs most effectively with the involvement of microorganisms. Many groups of microorganisms, both aerobic and anaerobic, are capable of metabolizing the hydrocarbons contained in organic pollutants. Bacteria are the most active factors of crude oil degradation and act as the main factors degrading crude oil spills in the environment [5]. Among bacterial genera isolated from soils contaminated with petroleum substances, which show the ability to biodegrade hydrocarbons, we could mention: *Gordonia*, *Brevibacterium*, *Aeromicrobium*, *Dietzia*, *Burkholderia*, and *Mycobacterium* [6]. On the other hand, fungal genera, which are capable of the decomposition of hydrocarbons, include, among others: *Amorphoteca*, *Neosartorya*, *Talaromyces* and *Graphium* and yeast, i.e., *Candida*, *Yarrowia* and *Pichia* [7]. However, the biodegradation of oil-derived substances is more complex. It is faster in the presence of adequate oxygen because of the activity of oxidative and hydrolytic enzymes, which are critical for decomposing the petroleum molecules [8].

However, the biodegradation of petroleum hydrocarbons in the soil may be restricted by many factors, such as autochthonous microflora, availability of biogenic compounds, temperature, humidity, availability of oxygen, physicochemical properties of the soil, and the type and properties of contamination [9]. The efficiency of biodegradation can be increased by applying various bioremediation techniques using living organisms, or their parts, metabolites, or enzymes capable of removing contaminants, as well as by supporting the growth of soil microflora, for example, by adding nutrients, emulsifiers, or surfactants that increase the bioavailability of petroleum-based substances to microorganisms [10,11]. In recent years, researchers have been especially interested in the biopreparations available on the market. One such preparation is Fyre-Zyme, produced by International Enzymes. It contains enzymes with the addition of surfactants, whose role is to facilitate the desorption of oil-derived substances in the soil and stimulate microorganisms to degrade the released products. Additionally, this preparation is enriched with simple sugars, amino acids, and additional growth factors [12]. Although the use of such biopreparations has many advantages, it also has many drawbacks. Before their introduction, biopreparations have to be revived to achieve full degradation activity. Moreover, before purchase, it is impossible to verify whether the microorganisms that are contained in the biopreparation are not antagonists of those naturally occurring in the soil subject to remediation [13].

Enzymatic activity is one of the best indicators of changes occurring in the soil under the influence of natural and anthropogenic factors [14–16]. The biological activity of soil that is defined by the enzymatic activity is a measure of soil fertility and productivity. Enzymatic tests provide quantitative information on the functional diversity of microbiological activity as well as the chemical processes, rate of mineralization, and accumulation of organic matter in the soil. Moreover, enzymes play an active role in the decomposition of organic matter released into the soil by plant vegetation, the formation and decomposition of soil humus, release and availability of mineral substances to plants, molecular nitrogen fixation, nitrification and denitrification, as well as in the detoxification of xenobiotics [17].

Hence, this study aimed to assess the influence of Fyre-Zyme preparation on selected chemical parameters and the activity of dehydrogenases and phosphatases of petrol-contaminated soil. Through this study, the following were attempted to be determined: (i) the effect of petrol on the properties of sandy soil, (ii) changes in the properties of soil uncontaminated with petroleum-derived substances under the influence of Fyre-Zyme preparation, and (iii) the possibility of using the enzymatic activity to assess the effectiveness of Fyre-Zyme preparation in the process of bioremediation of petrol-contaminated soil.

## 2. Materials and Methods

### 2.1. Experimental Design

The experiment was carried out on soil samples taken from the organic-humic level of rusty soils typical for the Agricultural Experimental Station of the West Pomeranian University of Technology

in Lipnik, Poland (53°24′ N, 14°28′ E). This soil is characterized by the granulometric composition of loamy sand and a pH of 6.43 in 1 M KCl. The samples were sifted through a 2-mm mesh sieve and divided into eight fractions each weighing 1 kg. Four of them were contaminated with petrol at the dose of 50 g kg$^{-1}$ dry matter (DM), while the remaining four were left uncontaminated. Petrol was purchased at one of the gas stations owned by PKN Orlen, Poland. The petrol-contaminated samples and the uncontaminated ones were then supplemented with a solution of the enzyme reagent Fyre-Zyme at a concentration of 6% in the following combinations: 0 (control), once at 40 cm$^3$ kg$^{-1}$ DM, twice at 20 cm$^3$ kg$^{-1}$ DM at two-week intervals, and four times at 10 cm$^3$ kg$^{-1}$ DM at weekly intervals. According to the data provided by the manufacturer, an aqueous solution of the enzyme preparation Fyre-Zyme with a concentration of 6% in a dose of 10% in relation to the weight of soil should be used. The application should be repeated at specific intervals. In the presented studies, we decided to check different methods of application of Fyre-Zyme preparation. Table 1 presents the experimental combinations. The maximum water holding capacity of the samples was brought up to 60%, and the samples were incubated in the dark at 20 °C. The method of determining the maximum water holding capacity of the soil consisted of determining the amount of water retained in the soil after slow humidification and in relation to the weight of the soil. The amount of water retained was determined from the difference in the mass of the sample after and before moisturization with water. After determining the maximum water capacity, the wet soil samples were dried at 105 °C to obtain a constant mass. The dried soil samples were supplied with a sufficient amount of water to bring them to 60% of the maximum water holding capacity. Every 2 days, the water losses were supplemented by controlling the soil mass with a scale. On days 1, 7, 14, 21, 28, and 56 of the experiment, the activities of dehydrogenases (DHA, EC 1.1.1), alkaline phosphatase (ALP, EC 3.1.3.1), and acid phosphatase (ACP, EC 3.1.3.2) were determined in all the experimental combinations, in three replications. In addition, on days 1 and 56 of the experiment, the total contents of organic carbon, total nitrogen, and total sulfur were determined, in three replications.

**Table 1.** Experimental combinations.

| Treatment | C | C1 × 40 | C2 × 20 | C4 × 10 | P | P1 × 40 | P2 × 20 | P4 × 10 |
|---|---|---|---|---|---|---|---|---|
| Petrol (g kg$^{-1}$ DM) | 0 | 0 | 0 | 0 | 50 | 50 | 50 | 50 |
| 6% solution of FZ (cm$^3$) | 0 | 1 × 40 | 2 × 20 | 4 × 10 | 0 | 1 × 40 | 2 × 20 | 4 × 10 |

DM—dry matter, FZ—Fyre-Zyme preparation.

*2.2. Determination of Soil Chemical Parameters*

All reagents used for soil chemical properties determination were purchased from the Sigma-Aldrich, Poznań, Poland).

Soil samples were air dried and further homogenization with ceramic mortar to pass through a 125-μm sieve. Organic carbon (C$_{org}$) in mineral soil samples was determined by wet oxidation soil organic matter—the Tiurin method. This method is based on organic carbon oxidized to CO$_2$ by a mixture of potassium dichromate and sulphuric (VI) acid. In the Tiurin method, a mixture of soil, 0.4 N K$_2$Cr$_2$O$_7$ and concentrated H$_2$SO$_4$ and catalyst (Ag$_2$SO$_4$) is boiled for 5 minutes—the organic carbon in the soil is oxidized to CO$_2$ by the dichromate, which itself is reduced. After cooling, the dichromate which has not been consumed to oxidise soil carbon is determined by titrating with Mohr's salt (ammonium iron(II) sulfate—(NH$_4$)Fe(SO$_4$)$_2$·6H$_2$O) with the N-phenylanthranilic acid (2-(C$_6$H$_5$NH)C$_6$H$_4$COOH) as indicators, with the determination carried out in triplicate [18]. Total nitrogen (N$_{tot}$) and sulphur (S$_{tot}$) contents were determined by means of the elementary analyzer COSTECH ECS 4010 with a Zero Blank Autosampler (Costech Analytical Technologies, Inc., Valencia, CA, USA). The prepared soil samples were weighed (15–20 mg) in tin capsules for solid samples and placed in the autosampler. The sample and tin capsule reacted with oxygen and combusted at temperatures of 1700–1800 °C and the samples were broken down into their elemental components, N$_2$ and SO$_2$. High performance copper wires absorbed the excess oxygen not used for sample

combustion. The gases flowed through the gas chromatographic (GC) separation column, which was kept at a constant temperature (±0.1 °C). As they were passed through the GC column, the gases were separated and were detected sequentially by the thermal conductivity detector (TCD). The TCD generated a signal, which is proportional to the amount of element in the sample. The EAS software compared the elemental peak to a known standard material (after calibration) and generated a report for each element on a weight basis [19].

## 2.3. Determination of Soil Enzyme Activity

All reagents used for soil biochemical properties determination were purchased from the Sigma-Aldrich, Poznań, Poland). Activity of soil enzymes were measured by spectrophotometric methods, using UV-1800 spectrophotometer (Shimadzu Corporation, Kyoto, Japan).

The activity of DHA was determined spectrophotometrically according to the method by Casida et al. [20]. This method involves the incubation of soil with a colorless, water-soluble substrate, 2,3,5-triphenyltetrazolium chloride (TTC) for 24 h at 25 °C. TTC is enzymatically reduced to a colored, water-insoluble product, triphenylformazan (TPF). After incubation, TPF is extracted from the soil with ethanol and analyzed spectrophotometrically at a wavelength of 485 nm. The dehydrogenase activity was determined from the standard curve and expressed in mg TPF $kg^{-1}$ DM $h^{-1}$.

The activities of ALP and ACP were analyzed according to the method described by Tabatabai and Bremener [21]. Following the addition of disodium *p*-nitrophenyl phosphate hexahydrate solution in buffer at pH 11 for ALP and 6.5 for ACP, the samples were incubated for 1 h at 37 °C. The resulting *p*-nitrophenol (*p*-NP) was extracted, stained with NaOH, and then determined spectrophotometrically at 400 nm. The phosphatase activity was determined from the standard curve and expressed in mg *p*-NP $kg^{-1}$ DM $h^{-1}$.

## 2.4. Data Analysis

The obtained results were converted according to the formulas given by Kaczyńska et al. [22] and presented as the petrol impact index ($IF_P$) and the Fyre-Zyme preparation impact index ($IF_{FZ}$) on the determined parameters:

$$IF_P = \frac{A_P}{A_0}$$

$$IF_{FZ} = \frac{A_{FZ}}{A}$$

where $IF_P$ is the petrol impact index, $IF_{FZ}$ is the Fyre-Zyme impact index, $A_P$ is the value of the determined parameter in the petrol-contaminated soil, $A_0$ is the value of the determined parameter in the uncontaminated soil, and $A_{FZ}$ is the value of the determined parameter in the soil treated with Fyre-Zyme, $A$ is the value of the determined parameter in the petrol-contaminated and uncontaminated soils.

The calculated values of $IF_P$ and $IF_{FZ}$ were analyzed by two-factor variance by considering the dose of petrol and the method of Fyre-Zyme application as variable factors. Then, the mean results were compared using the post hoc Tukey honest significant difference test at the significance level of $p = 0.05$. The analyses were performed independently for each measurement date.

To determine which of the variable factors had the greatest influence on the calculated ratios of petrol and Fyre-Zyme preparation interactions, $\eta^2$ analysis was carried out. This analysis describes the ratio of variance of the dependent variable explained by the independent variable predictor [23].

Additionally, the predictable efficiency of the $IF_{FZ}$ index in the assessment of the effect of Fyre-Zyme preparation on the soil enzyme activity was determined using sensitivity, specificity, and the area under the receiver operating characteristic (ROC) curve (AUC): 0.9–1 = excellent; 0.8–0.9 = good; 0.7–0.8 = fair;

0.6–0.7 = poor; < 0.6 = fail [24]. The ROC curves were constructed by calculating sensitivity and specificity using the following equations [25]:

$$sensitivity = \frac{TP}{TP + FN}$$

$$specifity = \frac{TN}{TN + FP}$$

where *TP* is a true positive, *TN* is a true negative, *FP* is a false positive (type I error), and *FN* is a false negative result (type II error). The ROC value, which is a scalar measure of the anticipated discrimination, refers to the area under the curve that combines the *TP* and *FP* proportions for an infinite number of limit values [26].

## 3. Results and Discussion

Table 2 presents the values of the content of organic carbon ($C_{org}$), total nitrogen ($N_{tot}$), total sulfur ($S_{tot}$), and the ratio of C:N in the soil uncontaminated with petrol at the beginning of the experiment.

**Table 2.** Initial chemical properties of soil uncontaminated with petrol.

| $C_{org}$ (g kg$^{-1}$) | $N_{tot}$ (g kg$^{-1}$) | C:N | $S_{tot}$ (g kg$^{-1}$) |
|---|---|---|---|
| 5.99 ± 0.12 | 0.48 ± 0.01 | 12.48 ± 0.14 | 0.19 ± 0.01 |

$C_{org}$, organic carbon content; $N_{tot}$, total nitrogen content; C:N, carbon-to-nitrogen ratio $S_{tot}$, total sulfur content.

Soil contamination with petrol resulted in a slight increase in the average content of $N_{tot}$ ($IF_P = 1.080$) and $S_{tot}$ ($IF_P = 1.067$) and a decrease in the C:N ratio ($IF_P = 0.945$), while the $C_{org}$ content was close to that of the uncontaminated soil ($IF_P = 1.020$) (Figure 1).

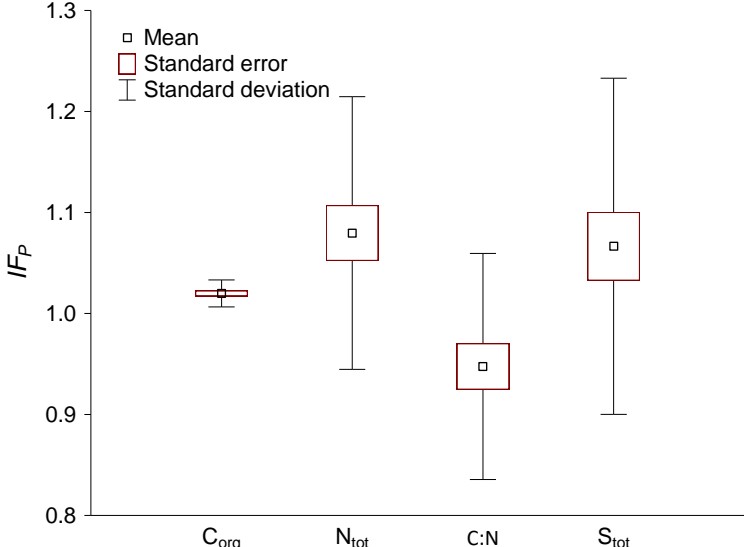

**Figure 1.** Mean values of petrol effect ($IF_P$) on chemical properties in soil uncontaminated with petrol; $C_{org}$, organic carbon content, $N_{tot}$, total nitrogen content, C:N, carbon-to-nitrogen ratio, $S_{tot}$, total sulfur content.

Many authors have demonstrated that the contamination of soil with oil-derived substances caused an increase in the $C_{org}$ content in the soil [27–30]. This is due to the presence of petroleum hydrocarbons in the oil, which may lead to the anthropogenic addition of carbon to the soil [31]. However, this effect is usually short-lived, as the content of light hydrocarbon fractions in the petrol

can quickly escape from the soil [32]. Therefore, it was not observed in the present study. Moreover, Schreier et al. [33] reported that organic carbon content may indicate the hydrocarbon contamination of soil. The changes in the soil organic carbon content may affect the balance of other soil nutrients, including the content of nitrogen, sulfur, phosphorus, or potassium, which may be used for the degradation of hydrocarbons present in the soil [28,30,34,35].

The application of Fyre-Zyme to both uncontaminated and petrol-contaminated soil increased the content of $C_{org}$ and $N_{tot}$, as evidenced by the $IF_{FZ}$ values above 1 (Table 3). However, a comparison of the different applications of Fyre-Zyme preparation indicated that in soil uncontaminated with petrol, the best statistical effect occurred after adding 40 cm$^3$ kg$^{-1}$ DM at once (for $C_{org}$ on both measurement dates, and for $N_{tot}$ on day 56). On the other hand, in the soil contaminated with petrol, no significant differences were found in the interaction of different applications of the enzyme reagent. For $C_{org}$, the $IF_{FZ}$ value was found to be significantly lower than that of the other enzyme reagent applications only on day 1 following the addition of 4 × 10 cm$^3$ kg$^{-1}$ DM, whereas for $N_{tot}$, a significant increase in the $IF_{FZ}$ value was noted on day 56 after the addition of 4 × 10 cm$^3$ kg$^{-1}$ DM of the enzyme reagent. Comparison of $IF_{FZ}$ values for C:N ratio and $S_{tot}$ content did not show significant differences between different applications of the Fyre-Zyme preparation in the uncontaminated soil. On the other hand, a significantly lower $IF_{FZ}$ value was found for C:N ratio on day 56 and for $S_{tot}$ on day 1 in the petrol-contaminated soil after the application of 4 × 10 cm$^3$ kg$^{-1}$ DM of the enzyme reagent.

**Table 3.** Indices of Fyre-Zyme effect ($IF_{FZ}$) on chemical parameters in soil uncontaminated and contaminated with petrol.

| Day | Uncontaminated Soil | | | Soil Contaminated with Petrol | | |
|---|---|---|---|---|---|---|
| | 1 × 40 | 2 × 20 | 4 × 10 | 1 × 40 | 2 × 20 | 4 × 10 |
| | $C_{org}$ | | | | | |
| 1 | 1.111 ± 0.003 [a] | 1.001 ± 0.008 [d] | 1.059 ± 0.008 [b] | 1.094 ± 0.005 [a] | 1.095 ± 0.014 [a] | 1.031 ± 0.001 [c] |
| 56 | 1.144 ± 0.015 [a] | 1.022 ± 0.025 [b] | 1.023 ± 0.006 [b] | 1.128 ± 0.045 [a] | 1.148 ± 0.025 [a] | 1.164 ± 0.017 [a] |
| | $N_{tot}$ | | | | | |
| 1 | 1.082 ± 0.036 [a] | 1.058 ± 0.056 [a] | 1.093 ± 0.041 [a] | 1.093 ± 0.086 [a] | 1.065 ± 0.104 [a] | 1.095 ± 0.096 [a] |
| 56 | 1.217 ± 0.072 [a,b] | 1.023 ± 0.049 [c] | 1.027 ± 0.053 [c] | 1.063 ± 0.101 [b,c] | 1.109 ± 0.083 [b,c] | 1.336 ± 0.033 [a] |
| | C:N | | | | | |
| 1 | 1.028 ± 0.032 [a] | 0.947 ± 0.046 [a] | 0.969 ± 0.032 [a] | 1.004 ± 0.076 [a] | 1.035 ± 0.102 [a] | 0.947 ± 0.085 [a] |
| 56 | 0.942 ± 0.064 [a,b] | 1.001 ± 0.075 [a,b] | 0.998 ± 0.051 [a,b] | 1.065 ± 0.064 [a] | 1.039 ± 0.055 [a] | 0.871 ± 0.015 [b] |
| | $S_{tot}$ | | | | | |
| 1 | 1.042 ± 0.056 [b] | 1.019 ± 0.035 [b] | 1.065 ± 0.029 [b] | 1.222 ± 0.081 [a] | 1.088 ± 0.046 [a,b] | 1.023 ± 0.053 [b] |
| 56 | 1.090 ± 0.038 [a] | 1.088 ± 0.008 [a] | 1.027 ± 0.065 [a] | 0.876 ± 0.029 [b] | 0.885 ± 0.041 [b] | 0.889 ± 0.050 [b] |

Data are expressed as a mean ± SD of three replicates; the same letters (a, b, c) in line are assigned to the same homogeneous groups (Tukey HSD test) with decreasing values differing from each other at the level of $p = 0.05$; $C_{org}$, organic carbon content; $N_{tot}$, total nitrogen content; C:N, carbon-to-nitrogen ratio $S_{tot}$, total sulfur content.

A comparison of the results of $\eta^2$ analysis for the determined chemical parameters of the soil indicated that the formation of $C_{org}$ and $N_{tot}$ content were influenced by the dose of petrol as well as by the application method of the enzyme reagent Fyre-Zyme. However, for the C:N ratio, the petrol dose as well as the interaction between the petrol dose and the method of enzyme reagent application were found to have the greatest influence (Table 4). Fyre-Zyme is a rich source of simple sugars, amino acids, and various biogenic factors. Therefore, changes in the content of elements in the soil may occur after the introduction of this enzyme preparation [9]. Moreover, the study carried out by Krzysko-Łupicka et al. [36] revealed that the introduction of an aqueous solution of Fyre-Zyme enzyme preparation at a concentration of 6%, at a dose of 10% (w/w), to the soil contaminated with oil-derived substances increased the effectiveness of the hydrocarbon removal. In the initial period (after 6 hours), the degree of contaminant biodegradation was 29%, corresponding to the baseline and control sample content. After 30 and 60 days of incubation, 47 and 52% of the initial contaminants were degraded, respectively. However, during the incubation, spontaneous transformations of petroleum-derived

substances occurred simultaneously in the control samples due to the activity of the soil microflora. Taking this fact into account, the degree of biodegradation in the presence of Fyre-Zyme was higher by 23–24% compared to the control. Further studies revealed that the application of Fyre-Zyme preparation led to the greatest decrease in the content of n-alkanes with the number of C8–C16 carbon atoms—that is, those dominant in petrol [12]. This may cause changes in the C:N ratio. The influence of experimental factors on the content of $S_{tot}$ was quite different. For this parameter, the greatest influence was exerted by the interaction between the petrol dose and the measurement day and the interaction between all three determined parameters.

**Table 4.** Percentage share of observed variability factors $\eta^2$ on soil chemical parameters.

| Variable Factor | $C_{org}$ | $N_{tot}$ | C:N | $S_{tot}$ |
|---|---|---|---|---|
| Petrol (P) | 37.02 | 54.64 | 37.07 | 3.46 |
| Fyre-Zyme Dose (FZ) | 25.78 | 16.09 | 14.87 | 8.41 |
| Day of Experiment (D) | 12.46 | 0.43 | 13.57 | 7.72 |
| P × FZ | 9.02 | 9.96 | 16.42 | 6.17 |
| P × D | 9.55 | 5.58 | 0.34 | 34.09 |
| FZ × D | 1.24 | 1.97 | 2.45 | 12.87 |
| P × FZ × D | 4.32 | 10.58 | 13.04 | 26.12 |
| Error | 0.60 | 0.75 | 2.23 | 1.17 |

$C_{org}$, organic carbon content; $N_{tot}$, total nitrogen content; C:N, carbon-to-nitrogen ratio $S_{tot}$, total sulfur content.

A comparison of the activity of the determined enzymes in the soil not contaminated with petrol revealed no significant differences in the activity of DHA on subsequent measurement dates. In contrast, the ALP activity was statistically highest on day 7 and that of ACP on day 1 (Table 5). Soil contamination with petrol at the dose of 50 g kg$^{-1}$ DM resulted in the inhibition of ACP and ALP on all measurement dates—$IF_P$ values below 1 and the activation of DHA—$IF_P$ values above 1 (Figure 2). Previous studies indicated that the reaction of soil enzymes with oil-derived substances depends mainly on their type. Contamination of the soil with light crude oil fractions mainly stimulated the activity of oxidoreductases [3], but inhibited the activity of hydrolases [37], which was also confirmed by the results of the present study. On the other hand, the introduction of heavier crude oil fractions such as diesel oil, spent engine oil, or coal tar creosote into the soil negatively impacted the activity of all soil enzymes [9,17,31,38,39].

**Table 5.** Activity of soil enzymes in uncontaminated soil.

| Day | DHA (mg TPF kg$^{-1}$ DM h$^{-1}$) | ALP (mg $p$-NP kg$^{-1}$ DM h$^{-1}$) | ACP (mg $p$-NP kg$^{-1}$ DM h$^{-1}$) |
|---|---|---|---|
| 1 | 1.69 ± 0.04 [a] | 67.20 ± 6.65 [c] | 188.24 ± 10.21 [a] |
| 7 | 1.71 ± 0.06 [a] | 139.94 ± 10.64 [a] | 177.31 ± 8.30 [b] |
| 14 | 1.66 ± 0.15 [a] | 81.08 ± 5.89 [b] | 180.45 ± 6.83 [a,b] |
| 21 | 1.77 ± 0.07 [a] | 73.48 ± 6.76 [b,c] | 177.76 ± 2.91 [b] |
| 28 | 1.71 ± 0.02 [a] | 71.49 ± 3.55 [b,c] | 183.53 ± 11.79 [a,b] |
| 56 | 1.79 ± 0.08 [a] | 67.06 ± 3.98 [c] | 180.56 ± 2.18 [b] |

Data are expressed as a mean ± SD of three replicates; the same letters (a, b, c) in column are assigned to the same homogeneous groups (Tukey HSD test) with decreasing activity differing from each other at the level of $p$ = 0.05;; DHA dehydrogenases; ALP, alkaline phosphatase; ACP, acid phosphatase; TPF, triphenylformazan; $p$-NP, $p$-nitrophenol; DM, dry matter.

An analysis of the effect of Fyre-Zyme preparation ($IF_{FZ}$) on DHA in the uncontaminated and petrol contaminated soil, based on the values of the enzyme coefficients, indicated that the values were close to 1, on all measurement dates, for each type of preparation application. This suggests small changes in the activity of this group of enzymes. However, a comparison of the $IF_{FZ}$ values for particular types of application of the enzyme preparation indicated that in the soil not contaminated with petrol, only on days 1, 7 and 21, the values were significantly higher after the application of

$4 \times 10$ cm$^3$ kg$^{-1}$ DM than after the application of $1 \times 40$ cm$^3$ kg$^{-1}$ DM. On the other hand, in the soil containing petrol, significantly higher $IF_{FZ}$ values were recorded on day 21 after the application of $4 \times 10$, and on days 1 and 21 after the application of $2 \times 20$ cm$^3$ kg$^{-1}$ DM of the enzyme reagent, than after the application of $1 \times 40$ cm$^3$ kg$^{-1}$ DM (Table 6). Kaczyńska et al. [22] reported that DHA is one of the most important enzyme groups because it is present in all living cells of microorganisms. Hence, it is often considered as an indicator of the overall microbiological activity of the soil. This group of enzymes performs biological oxidation of organic matter in soil through hydrogen transfer from the organic medium to inorganic acceptors [40]. Many authors have also stated that DHA acts as an indicator of the rate of changes taking place in the soil [16,22,41–44].

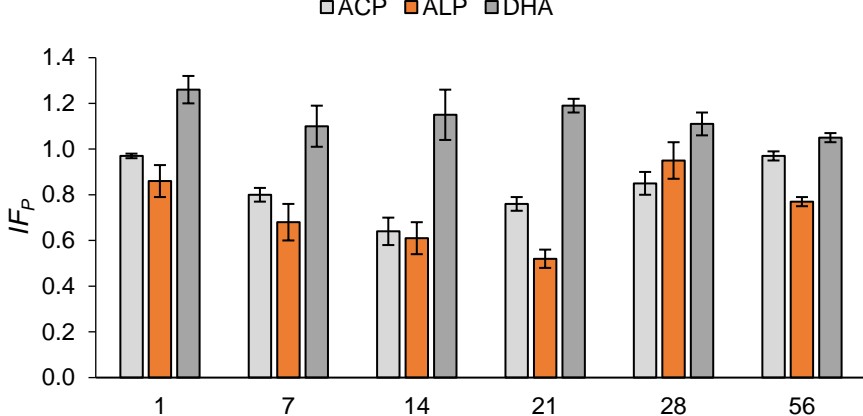

**Figure 2.** Values of petrol effect ($IF_P$) on enzyme activities in soil uncontaminated with petrol; ACP—acid phosphatase, ALP—alkaline phosphatase, DHA—dehydrogenases.

**Table 6.** Indices of Fyre-Zyme impact ($IF_{FZ}$) on soil biochemical parameters.

| Day | Uncontaminated Soil | | | Soil Contaminated with Petrol | | |
|---|---|---|---|---|---|---|
| | $1 \times 40$ | $2 \times 20$ | $4 \times 10$ | $1 \times 40$ | $2 \times 20$ | $4 \times 10$ |
| | Dehydrogenases (DHA) | | | | | |
| 1 | 0.918 ± 0.043 [b,c] | 0.967 ± 0.041 [a,b] | 1.051 ± 0.039 [a] | 0.839 ± 0.045 [c] | 0.978 ± 0.036 [a,b] | 0.876 ± 0.013 [b,c] |
| 7 | 0.806 ± 0.052 [b] | 0.936 ± 0.034 [a,b] | 0.990 ± 0.068 [a] | 0.929 ± 0.056 [a,b] | 1.052 ± 0.045 [a] | 1.002 ± 0.055 [a] |
| 14 | 0.951 ± 0.089 [a] | 1.005 ± 0.059 [a] | 1.041 ± 0.098 [a] | 0.916 ± 0.029 [a] | 1.020 ± 0.032 [a] | 1.002 ± 0.051 [a] |
| 21 | 0.926 ± 0.044 [b] | 0.984 ± 0.020 [a,b] | 1.010 ± 0.012 [a] | 0.805 ± 0.018 [c] | 0.954 ± 0.022 [a,b] | 0.905 ± 0.041 [b] |
| 28 | 0.951 ± 0.013 [a] | 1.004 ± 0.009 [a] | 1.039 ± 0.037 [a] | 0.909 ± 0.069 [a] | 1.039 ± 0.053 [a] | 1.021 ± 0.079 [a] |
| 56 | 0.997 ± 0.042 [a] | 0.987 ± 0.026 [a] | 0.926 ± 0.086 [a] | 0.935 ± 0.046 [a] | 1.013 ± 0.006 [a] | 1.040 ± 0.033 [a] |
| | Alkaline Phosphatase (ALP) | | | | | |
| 1 | 0.960 ± 0.091 [a] | 1.072 ± 0.102 [a] | 0.829 ± 0.081 [a] | 1.069 ± 0.103 [a] | 1.108 ± 0.066 [a] | 1.315 ± 0.114 [a] |
| 7 | 0.811 ± 0.076 [a] | 1.112 ± 0.109 [a] | 0.984 ± 0.038 [a] | 0.963 ± 0.093 [a] | 1.093 ± 0.099 [a] | 1.024 ± 0.087 [a] |
| 14 | 1.106 ± 0.101 [b] | 1.262 ± 0.114 [b] | 1.073 ± 0.098 [b] | 0.951 ± 0.056 [b] | 1.665 ± 0.124 [a] | 1.728 ± 0.132 [a] |
| 21 | 0.682 ± 0.056 [c] | 0.821 ± 0.078 [b,c] | 0.829 ± 0.081 [b,c] | 1.440 ± 0.129 [a,b,c] | 2.013 ± 0.187 [a] | 1.507 ± 0.091 [a,b] |
| 28 | 1.109 ± 0.059 [a] | 1.085 ± 0.104 [a] | 1.109 ± 0.102 [a] | 1.174 ± 0.027 [a] | 1.193 ± 0.041 [a] | 1.279 ± 0.061 [a] |
| 56 | 1.239 ± 0.106 [a,b] | 1.487 ± 0.123 [a] | 1.302 ± 0.107 [a,b] | 1.021 ± 0.023 [b] | 1.303 ± 0.099 [a,b] | 1.215 ± 0.111 [a,b] |
| | Acid Phosphatase (ACP) | | | | | |
| 1 | 0.674 ± 0.032 [c] | 0.992 ± 0.068 [a,b] | 1.137 ± 0.067 [a] | 0.683 ± 0.053 [c] | 0.973 ± 0.010 [b] | 1.057 ± 0.067 [a,b] |
| 7 | 0.607 ± 0.045 [d] | 1.014 ± 0.087 [b] | 1.178 ± 0.026 [b] | 0.812 ± 0.044 [c] | 1.118 ± 0.059 [b] | 1.352 ± 0.047 [a] |
| 14 | 0.742 ± 0.072 [d] | 0.958 ± 0.038 [c] | 1.142 ± 0.026 [b] | 0.836 ± 0.076 [c,d] | 1.321 ± 0.086 [a] | 1.371 ± 0.066 [a] |
| 21 | 0.809 ± 0.032 [e] | 1.052 ± 0.021 [c,d] | 1.193 ± 0.032 [a,b] | 0.945 ± 0.041 [d] | 1.103 ± 0.075 [b,c] | 1.241 ± 0.036 [a] |
| 28 | 0.928 ± 0.089 [a] | 0.982 ± 0.039 [a] | 1.071 ± 0.102 [a] | 0.991 ± 0.046 [a] | 1.108 ± 0.076 [a] | 1.086 ± 0.026 [a] |
| 56 | 1.004 ± 0.028 [a] | 1.014 ± 0.018 [a] | 1.032 ± 0.044 [a] | 1.018 ± 0.046 [a] | 1.002 ± 0.039 [a] | 1.035 ± 0.075 [a] |

Data are expressed as a mean ± SD of three replicates; the same letters (a, b, c, d, e) in line are assigned to the same homogeneous groups (Tukey HSD test) with decreasing values differing from each other at the level of $p = 0.05$.

In the uncontaminated soil, no significant differences were noted in the effect of different ways of enzyme reagent application on all the measurement dates (Table 6). A different effect was observed in

the soil contaminated with petrol. The highest $IF_{FZ}$ values were recorded on day 21. Moreover, on day 14 after the application of the enzyme reagent at doses of $4 \times 10$ and $2 \times 20$ cm$^3$ kg$^{-1}$ DM, the $IF_{FZ}$ values were significantly higher than after the application of $1 \times 40$ cm$^3$ kg$^{-1}$ DM. For ACP, the $IF_{FZ}$ values decreased significantly after the application of the enzyme reagent at a dose of $1 \times 40$ cm$^3$ kg$^{-1}$ DM, especially on the first measurement dates. However, they increased during the course of the experiment and were close to 1 on the last measurement date. On the other hand, from day 1 to day 21, after the introduction of the enzyme reagent at the doses of $4 \times 10$ and $2 \times 20$ cm$^3$ kg$^{-1}$ DM in both the uncontaminated and petrol-contaminated soil, the $IF_{FZ}$ values of ACP were found to be significantly higher than that after the application of $1 \times 40$ cm$^3$ kg$^{-1}$ DM. The highest $IF_{FZ}$ values of ACP were recorded in the petrol-contaminated soil on day 14 after the addition of the enzyme reagent at a dose of $2 \times 20$ cm$^3$ kg$^{-1}$ DM (1.321) and at a dose of $4 \times 10$ cm$^3$ kg$^{-1}$ DM (1.371). ACP and ALP play an important role in plant nutrition because they catalyze the hydrolysis of organic phosphorus, resulting in the formation of inorganic P that can be taken up by plants [45]. Phosphatases are ubiquitous in the soil and produced by microorganisms in response to low levels of inorganic phosphates. The amount of P available for plants in the soil is small, constituting only about 1–5% of the total P content [43]. Many authors have suggested that the examination of soil phosphatase activity may indicate the ability of soil to maintain its quality [14,44,46–49].

The introduction of Fyre-Zyme to the soil caused both inhibition and stimulation of the determined enzyme activities. The preparation contains enzymes, biosurfactants and nutrients. The results of the studies carried out so far on the influence of various additives on biochemical processes in the soil are also inconclusive. Both increases and decreases in degradation processes as a result of the introduction of biopreparations and different additives have been observed [3,11,22,39,50]. The effectiveness of this treatment is determined by numerous factors of a biological and physicochemical nature [51]. It is determined by the type of surfactant, amount and type of contamination and toxicity. The ability to adsorb on the soil matrix of both the pollutant and surfactant is also important. Moreover, the chemical composition and other environmental factors such as pH and temperature have an influence on the biodegradation process and enzymatic activity. The decrease in enzyme activity observed in the initial period of the experiment, especially DHA, may result from disturbances in soil air conditions [39]. It is also worth noting the quantitative and qualitative composition of indigenous microflora. The ability of this microflora to use hydrocarbons as a food substrate is important. Not without significance is also the possibility for microorganisms to use the surfactant or organic compounds introduced from outside as a food substrate. If the microorganisms are more likely to biodegrade the surfactant than the contaminant, the effectiveness of the surfactant decreases, which may be associated with a decrease in soil enzymatic activity [52]. In this case, the surfactant may repress the transcription/translation of enzymes required for the microbial metabolism and catabolism of the contaminants, and promote the development of bacterial populations that do not degrade the contaminant [53].

A comparison of the results of $\eta^2$ analysis performed for the determined soil enzymatic parameters showed that the petrol dose greatly influenced the activity of all the determined enzymes (Table 7). Additionally, for ALP, the measurement date had a similar influence as the petrol dose, whereas for ACP, the method of application of the enzyme reagent was influential. As mentioned above, Fyre-Zyme is a rich source of simple sugars, amino acids, and various biogenic factors. Therefore, the introduction of this enzyme preparation increases the source of substrates for enzymatic reactions in soil [9]. However, the addition of surfactants to the biopreparations may change the water–air interactions in the soil, which affects the enzymatic activity [54]. Moreover, many preparations supporting the degradation of oil derivatives contain mono- and dioxygenases, which were not analyzed in the present study [55].

A comparison of the ROC curves showed that for all the determined soil enzymes, the value of $p < 0.05$ was found for the application of the enzyme reagent at the dose of $2 \times 20$ cm$^3$ kg$^{-1}$ DM (Figure 3). For ALP, additionally, the value of $p < 0.05$ was found after the introduction of the enzyme reagent at the dose of $4 \times 10$ cm$^3$ kg$^{-1}$ DM. Considering the cut-off values, it can be concluded that the $IF_{FZ}$ values above 1.015 may indicate a positive influence of the enzyme reagent on the activity of

the determined enzymes (but not always). However, assuming that the diagnostic power of the ROC curves is expressed by the AUC values above 0.80, it can be concluded that only the application of the enzyme reagent at a dose of $4 \times 10 \text{ cm}^3 \text{ kg}^{-1}$ DM positively affected ALP (Table 8).

**Table 7.** Percentage share of observed variability factors $\eta^2$ on soil enzyme activities.

| Variable Factor | ACP | ALP | DHA |
|---|---|---|---|
| Petrol (P) | 41.76 | 38.73 | 79.97 |
| Fyre-Zyme Dose (FZ) | 42.22 | 6.10 | 13.13 |
| Day of Experiment (D) | 7.07 | 42.16 | 2.05 |
| P × FZ | 1.79 | 2.51 | 1.94 |
| P × D | 3.43 | 7.95 | 1.27 |
| FZ × D | 3.23 | 1.04 | 0.50 |
| P × FZ × D | 0.40 | 1.17 | 0.89 |
| Error | 0.11 | 0.34 | 0.25 |

ACP, acid phosphatase; ALP, alkaline phosphatase; DHA, dehydrogenases.

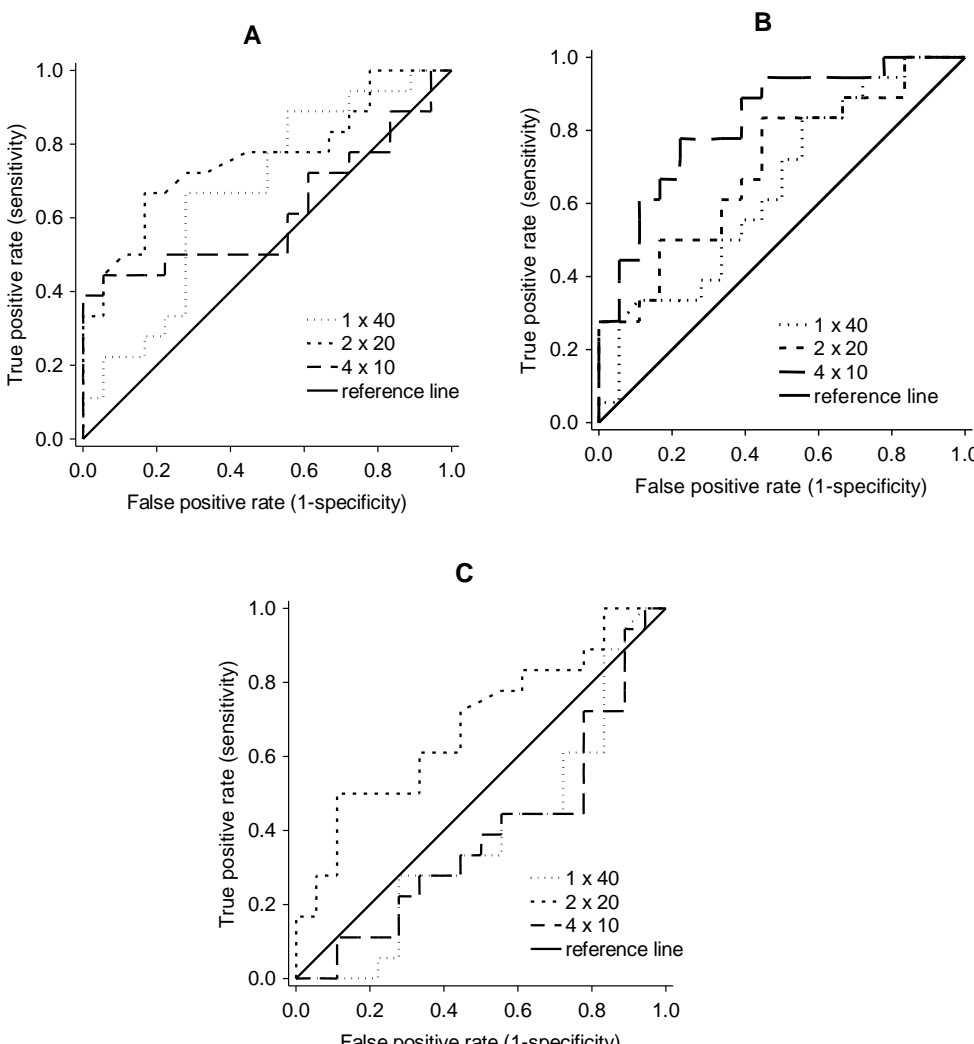

**Figure 3.** Receiver operating characteristic (ROC) curves for indices of Fyre-Zyme reagent impact on antivity of acid phosphatase (**A**), alkaline phosphatase (**B**), and dehydrogenases (**C**).

**Table 8.** ROC values of indices of Fyre-Zyme reagent impact on soil enzyme antivities.

| Application of Fyre-Zyme | Cut-Off Value | AUC | SE | *p* |
|---|---|---|---|---|
| Acid phosphatase (ACP) | | | | |
| $1 \times 40$ | 0.863 | 0.673 | 0.091 | 0.058 |
| $2 \times 20$ | 1.054 | 0.762 | 0.081 | 0.001 |
| $4 \times 10$ | 1.275 | 0.617 | 0.098 | 0.233 |
| Alkaline Phosphatase (ALP) | | | | |
| $1 \times 40$ | 0.941 | 0.644 | 0.093 | 0.121 |
| $2 \times 20$ | 1.130 | 0.704 | 0.057 | 0.019 |
| $4 \times 10$ | 1.214 | 0.872 | 0.069 | 0.001 |
| Dehydrogenases (DHA) | | | | |
| $1 \times 40$ | 0.807 | 0.387 | 0.096 | 0.241 |
| $2 \times 20$ | 1.015 | 0.685 | 0.089 | 0.038 |
| $4 \times 10$ | 0.877 | 0.389 | 0.096 | 0.247 |

AUC, area under the ROC curve; SE, standard error; *p*, level of probability.

Until now, there have been few reports in the literature about the effectiveness and mechanism of action of Fyre-Zyme in biodegradation processes. Krzysko-Łupicka et al. [36] reported that this preparation used in laboratory tests intensified the process of the removal of petroleum substances as early as several hours after its introduction to contaminated soil; it did not affect the loss of monoaromatic hydrocarbons but accelerated the degradation of PAHs and *n*-aliphatic hydrocarbons compared to the control. A new aspect of our study is also the determination of the side-effects of Fyre-Zyme preparation on activity of enzymes, which are one of the best indicators of soil ecochemical status.

## 4. Conclusions

Contamination of loamy sand with petrol caused slight changes in the $C_{org}$, $N_{tot}$, $S_{tot}$, and C:N ratio, and also stimulated the activity of DHA but inhibited the activity of ACP and ALP. The introduction of the enzyme reagent into the uncontaminated and contaminated soil increased the content of $C_{org}$ and $N_{tot}$. However, among the determined enzymes, the greatest changes were observed in phosphatase activity. The obtained results and ROC curves reveal that the application of the enzyme reagent at the dose of $4 \times 10$ cm$^3$ kg$^{-1}$ DM was the most beneficial in soil contaminated with petrol at the dose of 50 g kg$^{-1}$ DM, but it was the petrol content that mainly influenced the determined parameters of loamy sand.

**Author Contributions:** Conceptualization, K.C. (Kornel Curyło) and A.T.; methodology K.C. (Kornel Curyło), A.T. and G.J.;formal analysis, K.C. (Kornel Curyło), A.T. and G.J.; investigation, K.C. (Kornel Curyło), G.J., T.K.-Ł. and K.C. (Krystyna Cybulska); writing—original draft preparation, K.C. (Kornel Curyło); writing—review and editing, K.C. (Kornel Curyło), A.T. and G.J.; supervision, A.T. All authors have read and agreed to the published version of the manuscript.

**Funding:** This research and APC was funded by Subsidy of Polish Ministry of Science and Higher Education for West Pomeranian University of Technology in Szczecin number 503-07-083-08/04.

**Conflicts of Interest:** The authors declare no conflict of interest.

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
