# Peer review of "Analysis of Chemical and Biochemical Parameters of Petrol-Contaminated Soil after Biostimulation with an Enzyme Reagent"

_processes, doi:10.3390/pr8080949_

Round 1

Reviewer 1 Report

This paper investigates the effects of petrol and the enzyme reagent Fyre Zyme on selected chemical and biochemical properties of loamy sand. The results show that the application of the enzyme reagent at the application of 1*40 cm3/kg was the most beneficial. The research topic is interesting and has applications to the practical field. This reviewer has some comments or clarification questions that may help the authors to further improve the quality of the manuscript. After the comments are well addressed, the paper can be accepted for publication in Processes. The general and detailed comments are summarised as follows.

  1. In the test design, the total contents of carbon, total nitrogen and total sulfur are determined. How about the nutrient contents that are extractable by plants?
  2. In the contaminated samples, why are once 40 cm3/kg DM, twice 20 cm3/kg DM at two-week intervals and four times 10 cm3/kg DM at weekly intervals adopted? Please also explain the rational of using a concentration of 6% enzyme reagent.
  3. The maximum water holding capacity of the samples is brought up to 60%. Please clarify how the water holding capacity is measured.
  4. Please explain the basic functions of the dehydrogenases (DHA, EC 1.1.1), alkaline 97 phosphatase (ALP, EC 3.1.3.1), and acid phosphatase (ACP, EC 3.1.3.2) in soil science?
  5. It is concluded that the application of 1*40 cm3/kg was the most beneficial. This is for the soil samples contaminated with petrol at the dose of 50 g g·kg–1 dry matter (DM). If the soil samples are contaminated at other doses of petrol, does the obtained conclusion still apply? If not, this is the limitation of the study. The authors should state in the manuscript.

Author Response

1. In the test design, the total contents of carbon, total nitrogen and total sulfur are determined. How about the nutrient contents that are extractable by plants?

In the presented studies, the authors have determined only the contents of organic carbon, total nitrogen and total sulfur. The aim was to determine whether the Fyre-Zyme preparation affects the basic chemical properties of soil. In subsequent studies, the authors plan to extend the study to include the determination of the content of more elements in the total and soluble forms.

2. In the contaminated samples, why are once 40 cm3/kg DM, twice 20 cm3/kg DM at two-week intervals and four times 10 cm3/kg DM at weekly intervals adopted? Please also explain the rational of using a concentration of 6% enzyme reagent.

The authors explained in the text of the manuscript why they used an aqueous solution of the preparation at 6% concentration (manufacturer's recommendations). Moreover, In the presented studies authors decided to check different methods of application of Fyre-Zyme preparation (currently lines 101-105).

3. The maximum water holding capacity of the samples is brought up to 60%. Please clarify how the water holding capacity is measured.

The authors in the text of the manuscript described the method of measuring maximum water holding capacity (currently lines 107-114).

4. Please explain the basic functions of the dehydrogenases (DHA, EC 1.1.1), alkaline phosphatase (ALP, EC 3.1.3.1), and acid phosphatase (ACP, EC 3.1.3.2) in soil science?

The basic functions and significance of dehydrogenases and phosphatases in soil are given in the chapter "Results and discussion" in lines 248-253 (currently 274-279) and 271-277 (currently 295-301), respectively.

5. It is concluded that the application of 1*40 cm3/kg was the most beneficial. This is for the soil samples contaminated with petrol at the dose of 50 g g·kg–1 dry matter (DM). If the soil samples are contaminated at other doses of petrol, does the obtained conclusion still apply? If not, this is the limitation of the study. The authors should state in the manuscript.

The authors used only one dose of petrol in the presented studies. This is very high soil contamination. In the Conclusion chapter, the dose of petrol is given.

Reviewer 2 Report

General opinion

The manuscript is well prepared. Methodology and analysis of results don't raise any objections. However, there are some errors in describing results presented in tables (listed below). My suggestion is also to expand discussion (especially with regard to soil enzymes) and to emphasize more the novelty and usefulness of the results.

Detailed remarks:

Line 91 – mistake in the unit – it should be 50 g kg-1

Lines 92-94 – why such dose of petroleum and why such doses of Fyre-Zyme? It should be explained.

Where the petrol came from?

In how many repetitions the chemical and biochemical analysis were done?

Line 167 – unnecessary “,” before [29]

Lines 183-185 – the description in the text is not true according to the data in Table 3. In Table 3 a significantly lower IFFZ value was found for C:N ratio on day 56, not on day 1 as you wrote, and then for Stot the significant difference is not on day 56, but on day 1

Line 238 – that is not true - according to the Table 6 it is not on day 14, but on day 21 or on day 7.

Line 240 – there is also statistically significant difference on day 1 between 1x40 and 2x20. Why you did not mentioned that?

Lines 251-252 – How do you know that on day 21 it was significantly lower? According to Table 6 data and description below the table the same letters in line  (not in columns) are assigned to the same homogeneous groups

Lines 261 do 263 – that is not true for the day 28 (according to the data in Table 6 on day 28 there weren’t any significant differences for ACP). Please be precise. How do you explain such phenomenon – I mean differences between the way of adding enzyme reagent and their influence on ACP. In the discussion below this part there is nothing about that. This remarks is also valid for other enzymes.

Line 413 – mistake in reference

Author Response

1. Line 91 – mistake in the unit – it should be 50 g kg-1

The unit was changed (currently line 97)

2. Lines 92-94 – why such dose of petroleum and why such doses of Fyre-Zyme? It should be explained. Where the petrol came from?

The authors used only one dose of petrol in the presented studies. This is very high soil contamination. The authors used the same dose of petrol in previous studies. This manuscript is part of Kornel Curyło's doctorate. Therefore, no other petrol doses were introduced for comparison with previous results. The authors explained in the text of the manuscript why they used an aqueous solution of the preparation at 6% concentration (manufacturer's recommendations). Moreover, In the presented studies authors decided to check different methods of application of Fyre-Zyme preparation (currently lines 101-105). In the text of the manuscript, the authors stated where they obtained the petrol from (currently lines 97-98).

3. In how many repetitions the chemical and biochemical analysis were done?

Chemical and biochemical parameters were measuren in three replications. The authors gave this information in text of manuscript (currently lines 114-118).

4. Line 167 – unnecessary “,” before [29]

"," was deleted.

5. Lines 183-185 – the description in the text is not true according to the data in Table 3. In Table 3 a significantly lower IFFZ value was found for C:N ratio on day 56, not on day 1 as you wrote, and then for Stot the significant difference is not on day 56, but on day 1; Line 238 – that is not true - according to the Table 6 it is not on day 14, but on day 21 or on day 7; Line 240 – there is also statistically significant difference on day 1 between 1x40 and 2x20. Why you did not mentioned that?; Lines 251-252 – How do you know that on day 21 it was significantly lower? According to Table 6 data and description below the table the same letters in line  (not in columns) are assigned to the same homogeneous groups; Lines 261 do 263 – that is not true for the day 28 (according to the data in Table 6 on day 28 there weren’t any significant differences for ACP). Please be precise.

The authors thoroughly checked the description of the results and made all the corrections indicated.

6. How do you explain such phenomenon – I mean differences between the way of adding enzyme reagent and their influence on ACP. In the discussion below this part there is nothing about that. This remarks is also valid for other enzymes.

The authors tried to explain the observed changes in activity of the enzymes determined (currently lines 303-322).

7. Line 413 – mistake in reference

Reference was corrected (currently lines 491-492.

Moreover, the authors extended the discussion and tried to add information about the novelty of this research.

Reviewer 3 Report

Curyło and collegues assessed the effects of Fyre-Zyme application and petrol contamination on selected chemical (organic carbon, total nitrogen, total sulfur, C:N ratio) and biochemical (activity of dehydrogenases, acid and alkaline phosphatases) properties of loamy sand. In overall, manuscript has been written correctly. It needs some minor revisions before considering to be published. It would also be extremely interesting to extend the research to study changes in the soil microbiome before and after contamination, as well as after the use of preparations such as Fyre-Zyme. I encourage you to consider such a proposal, because it may increase the value of subsequent works of your team.

Minor suggestions:

  1. According to "Processes" template please include the corresponding author in the manuscript.
  2. Abstract is too long – it should be about 200 words maximum.
  3. Introduction – lines 45-50 - please add specific microorganisms capable of degrading hydrocarbons
  4. Introduction - If possible, please provide more information about Fyre-Zyme, e.g. what enzymes are contained in this preparation
  5. Materials and methods – 2.2. Determination of soil chemical parameters – please provide more information about used methodology.
  6. Page 9 - line 291 separates figure 3. and its caption.
  7. Please improve the discussion section.
  8. Please, indicate the novelty of your research compared to the previous research of your team.
  9. References 33. and 47. should be corrected.

Author Response

1. According to "Processes" template please include the corresponding author in the manuscript.

The authors added the email addresses of the affiliates and gave the author of the correspondence

2. Abstract is too long – it should be about 200 words maximum

The authors shortened the Abstract to 200 words

3. Introduction – lines 45-50 - please add specific microorganisms capable of degrading hydrocarbons

The authors added examples of genera of bacteria and fungi capable of biodegradation of hydrocarbons (currently lines 45-56)

4. Introduction - If possible, please provide more information about Fyre-Zyme, e.g. what enzymes are contained in this preparation

Fyre-Zyme from International Enzymes (according to the information provided by the manufacturer) is a proprietary formula of concentrated enzymes with the addition of biosurfactants, whose purpose is to facilitate the desorption of oil-derived particles in the soil and to stimulate soil bacteria to degrade released oil products. Fyre-Zyme is a rich source of simple sugars, amino acids and other growth factors. Therefore, the authors gave only information provided by the manufacturer. In further research, the authors plan to attempt to determine the composition of the preparation.

5. Materials and methods – 2.2. Determination of soil chemical parameters – please provide more information about used methodology.

The authors have extended the description of methods for determination of soil chemical properties.

6. Page 9 - line 291 separates figure 3. and its caption.

The authors have improved the formatting of the entire manuscript

7. Please improve the discussion section

The authors decided to leave the chapter "Results and Discussion" and not to split it into separate chapters "Results" and "Discussion". The authors believe that this makes the manuscript more readable

8. Please, indicate the novelty of your research compared to the previous research of your team.

The authors tried to add information about the novelty of their research.

9. References 33. and 47. should be corrected

References 33 and 47 (currently 36 and 54) were corrected

Moreover, the authors extended the discussion. In further research, the authors plan to study the biodiversity of bacteria and funghi by NGS method.